# The Role of Coinfections in the EBV–Host Broken Equilibrium

**DOI:** 10.3390/v13071399

**Published:** 2021-07-19

**Authors:** Yessica Sánchez-Ponce, Ezequiel M. Fuentes-Pananá

**Affiliations:** 1Research Unit in Virology and Cancer, Children’s Hospital of Mexico Federico Gómez, Mexico City 06720, Mexico; yes103_neutron@hotmail.com; 2Doctoral Program in Biomedical Science, National Autonomous University of Mexico, Mexico City 04510, Mexico

**Keywords:** EBV, HIV, beta herpesvirus, HPV, *P. falciparum*, *H. pylori*, periodontal bacteria, coinfection, immunosuppression, lymphomagenesis

## Abstract

The Epstein–Barr virus (EBV) is a well-adapted human virus, and its infection is exclusive to our species, generally beginning in the childhood and then persisting throughout the life of most of the affected adults. Although this infection generally remains asymptomatic, EBV can trigger life-threatening conditions under unclear circumstances. The EBV lifecycle is characterized by interactions with other viruses or bacteria, which increases the probability of awakening its pathobiont capacity. For instance, EBV infects B cells with the potential to alter the germinal center reaction (GCR)—an adaptive immune structure wherein mutagenic-driven processes take place. HIV- and *Plasmodium falciparum*-induced B cell hyperactivation also feeds the GCR. These agents, along with the B cell tropic KSHV, converge in the ontogeny of germinal center (GC) or post-GC lymphomas. EBV oral transmission facilitates interactions with local bacteria and HPV, thereby increasing the risk of periodontal diseases and head and neck carcinomas. It is less clear as to how EBV is localized in the stomach, but together with *Helicobacter pylori*, they are known to be responsible for gastric cancer. Perhaps this mechanism is reminiscent of the local inflammation that attracts different herpesviruses and enhances graft damage and chances of rejection in transplanted patients. In this review, we discussed the existing evidence suggestive of EBV possessing the potential to synergize or cooperate with these agents to trigger or worsen the disease.

## 1. Introduction

The Epstein–Barr virus (EBV) is a well-adapted human virus, and its infection is exclusive to our species, generally beginning in childhood and persisting throughout the life of most affected adults. This adaptation probably reflects at least a hundred thousand years of viral and human coevolution, helping shape an equilibrium wherein most people live with the virus without showing overt evidence of malaise. Nevertheless, this virus is often associated with severe diseases such as cancer and inflammatory and auto-immune disorders. Thus, EBV seems more of a pathobiont than a true pathogen, that is, an agent with the capacity to unleash its harmful potential in dysbiotic conditions [1]. The circumstances that lead to the disturbance of the viral-host balance are more often unknown. In this review, we focused on the existing evidence indicating that EBV-opportunistic mechanisms of pathogenicity can intersect with those of other infectious agents, resulting in detonation or worsening of the disease (Figure 1).

EBV’s main route of transmission is through saliva droplets. EBV first infects the lymphoepithelial tissues of the Waldeyer ring. EBV causes chronic infections oscillating between B cells and epithelial cells, the latter serving to amplify the number of infectious particles released into the saliva for efficient transmission to new hosts and, most likely, B cells providing the niche for persistence, particularly memory B cells [2,3]. Similar to that for the other members of the herpesvirus family, EBV oscillates between lytic and latent infectious cycles, perhaps with the former prevailing more in the epithelial cells and the latter more in the B cells, together assuring successful persistence at the population and individual levels.

EBV is found in latently infected cells with different active transcriptional programs, which are often referred to as latency programs. For instance, there is a latency III program in which the virus expresses the complete repertoire of latent proteins: the EBV nuclear antigens EBNA1; EBNA2; the family of EBNA 3A, 3B, and 3C; EBNA-LP (leader protein), and three latent membrane proteins LMP1, LMP2A, and LMP2B. The virus progressively switches off the expression of these proteins, beginning with the EBNAs in latency II and then the EBNAs and the LMPs in latency I, except for EBNA1 whose expression is maintained through the three transcriptional programs. EBNA1 serves to replicate the viral genome during the cell cycle and segregates viral genomes to both the daughter cells after mitosis, thus ensuring that the infection is not lost during cell replication. According to the germinal center (GC) model [2], these latency programs allow EBV to transit to the B cell differentiation pathway, changing from an activated to a memory state. Memory B cells are quiescent long-lived cells in which EBV turns off all viral protein expression (latency 0) [3]. Although EBV also expresses several non-coding RNAs throughout all latency programs, latency 0 is most likely the least harmful and the most prevalent one in asymptomatic carriers. The remaining latencies are active in the neoplasms associated with a viral infection and, in the case of EBV-induced B cell lymphomas, these neoplasms usually phenotypically correlate with an intermediate state of the GC reaction (GCR) [3].

## 2. EBV in Immunocompromised Individuals

The capacity of EBV to downregulate the viral protein expression also allows it to hide from antagonistic host immune responses. Presently, the precise mechanisms and immune populations that help maintain the EBV-host equilibrium are mostly unclear. Most likely, this is importantly aided by CD8 T cells, CD4 T cells, and natural killer (NK) cells [4], since one of the most evident mechanisms of a broken equilibrium is made evident by the improper functioning of these immune cells, as it occurs after primary immunodeficiencies (inborn errors of immunity) or because of pharmacological immunosuppression in the organ- or progenitor cell-transplanted patients. Similarly, individuals infected with the human immunodeficiency virus (HIV) often endure uncontrolled bursts of EBV expansion with the capacity to lead to different types of EBV-associated neoplasms. Before the introduction of antiretroviral therapy (ART), the so-called AIDS-defining cancers emerged within a very deteriorated host immune system that was unable to efficiently survey for virus-driven oncogenesis. These neoplasms are often polyclonal cellular expansions, as in Kaposi sarcoma virus (KSHV)-induced sarcomas, Castleman disease, and EBV lymphoproliferative disorders (LPD).

## 3. EBV and KSHV Coinfect B Cells in Primary Effusion Lymphoma (PEL) and Germinotropic LPD

One of the clearest and best-understood examples of EBV’s cooperation with other pathogens is with KSHV, a gammaherpesvirus similar to EBV. Between 60% and 90% of PEL cases harbor EBV and KSHV genomes coinfecting almost all lymphoma cells [5]. PEL is a neoplasm that almost exclusively surfaces in AIDS patients and is characterized by an accumulation of tumor cells in the serous cavities of the body, such as the pleura, pericardium, and peritoneum, although an extracavitary presentation has also been reported [6]. A different gene expression signature is observed in single-infected PELs when compared with that in coinfected PELs, suggesting distinct PEL subtypes depending on the presence or absence of EBV [6]. EBV-positive PELs exhibit evidence of somatic hypermutation, which is supportive of a post-GC B cell origin. On the contrary, EBV-negative PEL is rarely hypermutated, suggesting a pre-GC origin [7]. Indeed, PEL cells morphologically resemble plasmacytic/plasmablastic cells [8]. KSHV and EBV also coinfect the cells of germinotropic LPD [9]. Germinotropic LPDs (GLPD) morphologically resemble EBV-positive extracavitary PEL, but are mostly found in immunocompetent hosts. Since this LPD is newer and there are only a few reported cases in the literature, most studies addressing EBV and KSHV interaction mechanisms are always considered exclusive of PEL.

It is debatable as to which virus initiates the lymphomagenesis process; since KSHV is present in all PELs, it appears to be the main driving force. However, KSHV cannot transform B cells nor can it persistently infect them in vitro [10]. On the contrary, EBV efficiently immortalizes and activates B cells. Perhaps because of this reason, EBV facilitates the establishment of sustained KSHV infection of B cells, thus allowing KSHV higher loads per cell—an activity credited to EBNA1 [11,12]. Thus, it has been proposed that EBV precedes KSHV, thus rendering EBV-infected cells more susceptible to KSHV infection [7]. However, another study found that persistent infection of peripheral B cells with KSHV was achieved when the cells were exposed to KSHV 24 h before EBV infection [11]. Regardless of which virus infects first, both are believed to be necessary to sustain PEL, as together they provide selective advantages to the coinfected cells [12]. For instance, the coinfected cells present an increased expression of cellular genes involved in the anti-apoptotic processes, mitochondrial functions, and cell cycle, while showing a decreased expression of genes involved in innate immune responses and cytokine signaling [13].

The contribution of EBV and KSHV in the lymphomagenesis process does not strictly follow the canonical patterns of transformation established for each virus individually. It thus seems that EBV and KSHV must establish a delicate balance in coinfected cells, blending and co-regulating their transforming mechanisms. For example, EBV presents a restricted gene expression pattern with a latency I program in coinfected PEL cell lines [14] because the KSHV LANA protein inhibits the master promoter regulator of EBV latency III (the Cp). LANA binds the corepressor mSin3 and together recruits the methylase MeCP2-silencing Cp through methylation-mediated repression [15]. A bidirectional regulatory mechanism that promotes the maintenance of dual latency in coinfected PEL cells through the inhibition of lytic replication has also been proposed [16]. KSHV LANA and RTA proteins transactivate EBV LMP1, which in turn represses lytic replications of both viruses [17,18,19]. RTA and BZLF1 are the main transcription factors controlling the activation of the KSHV and EBV lytic cycles, respectively. Colocalization and coprecipitation experiments suggest that both proteins inhibit each other through direct interactions, which interferes with the formation of their respective multimeric active complexes [18,19,20]. Since EBV is in latency I, it is not clear when the LMP1-mediated lytic cycle suppression is active. Other studies support a loose control of EBV transcription with a sporadic expression of other genes, mainly LMP1 [14,21,22]. Enhanced expression of EBV lytic genes has also been observed in coinfected PEL cells, which can be attributed to an increased expression of the transcription factor BLIMP1 [23]. BLIMP1 is involved not only in the maintenance of PEL [24] but also in the induction of both EBV immediate early genes *BZLF1* and *BRLF1* [25]. The expression of BLIMP1 marks the end of the GCR, sending GC B cells into antibody-producing plasma cells, which correlates with EBV reactivation [26]. Similarly, the BLIMP1 expression correlates with epithelial cell differentiation and the induction of the EBV lytic program [25].

It remains uncertain how the lymphomagenesis process is enhanced through coinfection. At least for PEL, this is largely aided by the HIV-imposed immunosuppression and HIV potential lymphomagenesis mechanisms (see below). In addition to immunosuppression, it remains reasonable to believe that KSHV, with a limited transforming capacity, is assisted by EBV, whose B cell transforming capacity is notably superior [11]. However, EBV displays a restricted latency, without the expression of its canonical oncogenes. Although KSHV is also responsible for Kaposi sarcoma and multicentric Castleman disease, these are considered to be polyclonal cellular expansions rather than true neoplasms. It has also been proposed that EBV contributes to the lymphomagenesis process through the regulation of the *TCL1* oncogene, which also occurs in EBV-positive Burkitt lymphoma (BL), although PEL appears to occur in a TCL1-independent manner [27]. The *myc* translocation characteristic of BL is also absent in PEL. Nevertheless, the EBERs and EBNA1 are deemed necessary for the proliferation of coinfected PEL cells [21,28]. It is also possible that, at the beginning of infection and before their repression, EBNA2 and other EBV products promote early transcriptional conditions, which modulate the expression of KSHV and cellular genes that together stage the permissive conditions for KSHV-persistent infection and the emergence of PEL [11]. KSHV genes *LANA*, *vFLIP*, and *vIRF* interact with different tumor suppressors and inhibit the apoptotic processes [29,30,31]. In addition, LANA, together with LMP1, can independently induce the promoter of *UCH-L1* through the activation of RBP-Jk. UCH-L1 is an oncogenic protein detected in several cancers [32], including some of lymphoid origin [33,34]. In PELs with LANA and LMP1 co-expression, both proteins synergize to activate *UCH-L1*, which enhances a tumorigenic phenotype with increased proliferation, adhesion, cell migration, and inhibition of apoptosis [35].

Humanized mice models have also aided in the understanding of how infection with both viruses proceeds. These models support that lytic reactivation plays a central role in the development of PEL, as it was observed that a significant proportion of coinfected cells express lytic genes; this observation was also associated with greater efficiency of lymphomagenesis [13,36,37,38]. This event may seem contrary to the idea shaped in cell lines about the negative bidirectional regulation of the KSHV and EBV lytic cycles [16,19]. Lytic genes can mediate the local release of cellular and viral cytokines, such as KSHV vIL-6, vGPCR, K1 and K15, and EBV vIL-10 (BCFR1), v-Bcl-2 (BHRF1), etc., which can aid oncogenesis through the promotion of proliferation, cell survival, immunosuppression, neoangiogenesis, and activation of oncogenic signaling pathways, such as NF-κB [39,40,41,42,43] (Figure 2). Altogether, these studies shed light on the complex interactions between KSHV and EBV, which support a balanced mutual influence that facilitates the persistence of both viruses toward enhancing their tumorigenic potential.

## 4. EBV and Betaherpesviruses in the Transplanted Patient

The human betaherpesviruses cytomegalovirus (CMV) and the roseoloviruses human herpesvirus 6A (HHV6A), human herpesvirus 6B (HHV6B), and human herpesvirus 7 (HHV7) are also life-persistent common viruses that exert direct cytopathic effects on several organs. These viruses have been proposed as potential cofactors in the progression of different neoplasms [44,45,46], and they are particularly relevant in settings with inadequate immunosurveillance, such as in transplanted patients subjected to pharmacological immunosuppression [47,48]. In these patients, the detection of these viruses has been associated with mutual exacerbation of their infectious cycles, graft rejection, and the development of post-transplant LPD (PTLD) [49,50]. For these reasons, multiple studies have addressed viral loads of EBV and the betaherpesviruses in the peripheral blood of post-transplanted patients as a measure of infection or reactivation. These studies revealed a codetection rate of 2.6–32.7% [51], sometimes without clinical significance [51,52,53,54]. In other studies, however, viral codetection appeared to influence the clinical outcomes. For example, the codetection of EBV and CMV has been associated with graft damage in up to 24.5% of cases [51,55] and with graft rejection in up to 11.1% of cases [51,56]. In children with renal transplants, we observed that EBV and HHV7 codetection was associated with a 5.5-fold-enhanced risk of graft rejection, while that of triple EBV, HHV6, and HHV7 infection was associated with an increased rejection rate by 17.6-fold [57]. Concerning PTLDs, these are a heterogeneous group of lymphoid disorders ranging from indolent polyclonal proliferations to aggressive lymphomas [58]. Although one study did not observe any association between EBV and CMV codetection and PTLD [54], multiple other studies have documented that their codetection increased the risk of PTLD [55,59,60,61].

Unfortunately, all of the above-mentioned reports are associative, and it remains uncertain whether detectable viral loads are indicative of causality or whether a worsened disease influences viral loads. It is therefore important to mention that, although these viruses are present in most of the population, they are generally undetectable or detected in low viral loads in immunocompetent carriers. Moreover, although high viral loads play an important role in the diagnosis, prevention, and treatment of associated diseases, there seems to be a lack of standardization of the methods used to measure them and there is also no consensus on the viral load threshold beyond which they should be considered to be clinically relevant [62]. Different individual or progressing thresholds have been proposed to identify or define groups of transplanted patients with an enhanced risk to develop PTLD, ranging from 500 to 4000 EBV copies per μg or per mL of blood DNA [60,63,64]. The association of viral loads with the clinical outcomes is blurrier when multiple viruses are simultaneously detected. The predictive value of codetection loads is controversial and unclear, with some patients presenting with “high” viral loads and remaining stable for months or years and others presenting with “low” viral loads with significant clinical implications [55]. In the latter study, there was a greater amount of graft dysfunction in patients with CMV and EBV codetection, but with “low” EBV loads (1000–2000 copies/mL of blood) relative to those with a “high” copy number (>5000). It is therefore clear that CMV and EBV individually influence graft damage, graft rejection, and PTLD, since prophylactic treatment with gancyclovir, a nucleoside analog that inhibits the DNA polymerase of herpesviruses, reduces these complications [47]. For this reason, EBV and CMV are routinely followed in transplanted patients. Nonetheless, the sole simultaneous detection of these viruses neither helps elucidate the role played by each virus nor their mechanisms of cooperation.

There are also other scenarios wherein the codetection of multiple herpesviruses has been reported. For instance, EBV and CMV together were associated with more severe forms of infectious mononucleosis [65,66,67], hepatitis, and hemophagocytic lymphohistiocytosis [66,67]. In a study on endemic BL, CMV, and KSHV were detected through immunohistochemistry in the adjacent non-neoplastic tissues. The presence of CMV and KSHV was associated with a shift in the mutational landscape of the neoplasm, with a lower load of classical BL mutations and additional activity of TCF3 and an enhanced expression of EBV lytic cycle genes [68]. In further mechanistic studies, herpesviruses codetection was shown to influence the dynamics of immune cells, such as in the formation and activity of NK cells [69,70,71], as well as the expansion and activity of B cells [72]. These mechanisms may deter immune responses directed against these viruses, facilitating their persistency. Indeed, CMV reactivation has been demonstrated to induce immunosuppression, which predicts EBV reactivation and PTLD risk [60]. CMV and EBV codetection has also been associated with a decreased frequency of CD56^dim^NKG2A^pos^KIR^neg^ NK cells, thus contributing to suboptimal EBV control in immunosuppressed children with PTLD [59]. HHV6 or EBV single infection of peripheral blood mononuclear cells induces greater levels of IL-6, TNFα, and IL-1β than in coinfection [73]. On the other hand, infection with HHV6 has been shown to influence the in vitro reactivation of EBV-infected B cells [74] (Figure 3). Taken together with the results of the above-mentioned studies, it can be said that a mild inflammatory reaction to the grafted organ can become exacerbated if EBV or other betaherpesviruses-infected immune cells arrive at the site of inflammation.

## 5. HIV Steady B Cell Stimulation Creates a Lymphomagenesis Permissive Environment

ART introduction shifted the conditions that restrict the life expectancy of people living with HIV (PLWH). Presently, cancer remains one of the most serious threats facing PLHW, and lymphomas are the most common cause of PLWH death in developed nations [75]. EBV is still responsible for more than half of the cases of lymphomas arising in these individuals [76]. For instance, Simard et al. reported that, while the burden of AIDS-defining cancers decreased from 18% to 4.2% with ART, the risk of classical Hodgkin lymphoma (cHL) increased [77], and EBV remains responsible for close to 100% of all cHL cases [78]. Moreover, although DLBCL has significantly decreased with ART, it remains the most common lymphoma post-ART, particularly the immunoblastic subtype that also tends to be EBV positive [79]. A search of The Surveillance, Epidemiology, and End Results tissue repository revealed that EBV is responsible for up to 64% of the BL of PLWH in the USA [80], although other countries have documented lower frequencies. This result indicates that the intersection between EBV and HIV goes beyond the loss of T cells, indicating other levels of interaction between both pathogens. This point is also illustrated by the fact that EBV-associated lymphomas arousing pre-ART tended to be latency III, while in those with controlled HIV viremia and more robust T cells numbers it tended to be latency I or II [81]. Thus, lymphomas arising in PLWH can be divided between those still dependent on strong immunosuppression and those developing within relatively normal CD4 T-cell numbers (e.g., >200/µL), with the EBV-associated cHL, DLBCL, and BL as examples of the latter. There are excellent reviews describing all EBV-positive lymphomas in HIV-infected people [82].

There are four different subtypes of cHL, of which the one that occurs most frequently in PLWH is the mixed-cellularity subtype. Mixed-cellularity cHL in PLWH is extremely similar phenotypically to the cHL of HIV-negative patients, except for a larger numbers of lymphoma cells [83], and EBV expresses a latency II program in both. Likewise, BL in PLWH depends on *myc* translocations to the immunoglobulin heavy or light chain loci, while EBV presents with a latency I gene expression program [84]. One phenotypic exception is that EBV-positive DLBCL, cHL, and BL more often present with plasmablastic/plasmacytoid features in PLWH. HIV lymphomagenesis mechanisms have been proposed to explain the high frequency of lymphomas in PLWH, despite that HIV has never been detected to infect lymphoma cells [85]. These lymphomagenesis-indirect mechanisms include chronic B cell activation and enhanced mutagenesis resulting from the continuous entry into the GCR, as well as T-cell exhaustion and senescence that may facilitate the immunoescape of lymphoma cells. Although these mechanisms are not EBV-specific, they may directly intersect with EBV-infected B cells transiting the GCR, for instance, cooperating with the capacity of EBV to enhance cell survival and proliferation. GCR elicits both memory B cells and plasma cells, but, contrary to memory cells, plasma cells are short-lived, and EBV-latent genes strongly antagonize the formation of this cell population [86,87] (Figure 2).

HIV-positive individuals harbor B cells with an activated phenotype, which can be explained by direct antigenic or indirect heterologous/polyclonal activation of B cells. Evidence of B cell hyperactivation includes the increased expression of activation markers [88] and hypergammaglobulinemia in in vivo and ex vivo cultures of B cells derived from PLWH [89]. Hyperactivated B cells explain a post-GC cell originating in the EBV-positive lymphomas in PLWH and the plasmablastic/plasmacytoid phenotype. Low levels of HIV replication in the lymph nodes or lymphatic mucosal tissues may directly activate B cells through HIV antigens. In addition, several HIV proteins are secreted, accumulate, and persist in the lymph nodes, in which they may help create a B cell stimulatory microenvironment. Particularly, HIV p17, p120, and Tat induce B cell lymphomas in mice recombinant models [90,91], and p17 and gp120 accumulate in GC even in individuals with undetectable HIV viremia [92]. Tat can be captured by B cells in which it activates oncogenic pathways [93]. This is also true of p17, and CXCR2 acts as a receptor of p17 in B cells triggering the PI3K/Akt oncogenic pathway [94,95]. Moreover, HIV virions carrying the CD40 ligand (CD40L) on their surface have also been reported. CD40L is incorporated into the HIV envelope upon viral budding from infected CD4 T-cells [96]. CD40 ligand-receptor interactions are the most important signals to activate B cells, after antigens, and they are the most important survival signals in the GC [97]. CD40L-positive HIV virions can strongly mimic the activation conferred by the CD40L on follicular CD4 T cells (T_FH_) [98,99]. In this scenario of specific and heterologous activation, HIV probably sends B cells to the GCR in which highly mutagenic mechanisms occur, such as somatic hypermutation and isotype switching together with extensive proliferation [97]. Enhanced GCR probably amplifies the chance of off-target gene mutagenesis, such as the *myc*, *bcl2*, and *bcl6* translocations found in BL and DLBCL. Indeed, these lymphomas and cHL often lack the expression of antigen receptors (BCR) because of crippled mutations, which is illustrative of off-target mutagenesis. Receptor-less B cells should die of apoptosis because they lack the signals given by the tonic or antigen-driven BCR activity, but they may be rescued by EBV-latent genes [100,101]. Importantly, the CXCR2 expression is induced by EBV LMP1, a latency II protein expressed in EBV-positive cHL and some DLBCL [102]. EBV LMP1 mimics CD40 coreceptor signals [103], and thus CD40L also enhances the CXCR2 production by B cells, potentially creating a positive loop of B cell activation with HIV and/or secreted p17.

There are other potential mechanisms of HIV heterologous activation of B cells. Despite successful ART, PLWH harbor immune cells that constitutively secrete several different cytokines, chemokines, and other markers of inflammation. The elevated levels of inflammatory molecules contribute to HIV comorbidities, such as cardiovascular disease, neurocognitive impairment, and cancer [104]. The identity of the inflammatory molecules varies among studies, but they often point to IL-6, IL-10, TNFα, IL-1β, IP-10/CXCL10, and C reactive proteins, which may positively loop to further activate B cells. IL-6 has been shown to maintain GCR in murine models of persistent viral infection [105], while IL-10 has been shown to enhance B cell antibody production [106]. IL-6, IL-10, and TNFα are expressed by B cells upon CD40L activation [107], and enhanced expression of IL-6 and IgG has been documented after B cell interaction with CD40L-bearing HIV virions [98]. Indeed, the IL-6 levels do not decline even after years of successful ART [108,109], and the enhanced levels of IL-6 and other B cell stimulatory molecules precede lymphoma development [110,111]. The elevated levels of IL-6, IL-10, TNFα, and IL-1β have also been observed in the heterologous B cell stimulation mediated by p17, gp120, and Tat [90,91] (Figure 2).

Although hefty numbers of CD4 T cells have been observed in the periphery, a different scenario may be occurring in secondary lymph nodes and mucosal tissues. For instance, severe depletion of CCR5^pos^ CD4 T cells in the gastrointestinal (GI) tract has been documented [112,113], which is not restored upon ART [114,115]. This event may compromise the GI epithelial barrier, leading to increased translocation of microbial products into the lamina propria [116] as well as to severe dysfunction of the GI mucosal immune cells. For instance, GI Th17 and Th22 CD4 T cells are impaired, and the non-physiological levels of IL-22 and IL-17 contribute to the disruption of the homeostasis of the epithelial barrier [117,118,119]. Microbial translocation is considered an important contributor to the persistent inflammation and pathological activation of systemic immune cells, which altogether are responsible for the increased mortality of PLWH despite controlled viremia. B cells may also be targeted by this pathological chronic activation, increasing the risk of lymphoma. Among the most important translocated microbial products causing systemic activation of immune cells are TLR and NLRPR agonists and β-d-glucans [120,121].

Some studies suggest that cHL is more common among individuals on ART than in those without it [122]. cHL is also more common in individuals with heftier CD4 T cell numbers (between 200 and 500 cells/µL) [123], perhaps indicating a co-participative role of these cells. CD4 T cells are necessary for the GCR, particularly T_FH_. HIV and T_FH_ cells’ cooperative mechanisms have been described to affect the GCR function and the B cell terminal differentiation into memory and antibody-producing plasma cells. T_FH_ cells are expanded in PLWH correlating with increased numbers of GC B cells and plasma B cells and with gammaglobulinemia [124]. HIV replication in the lymph node T_FH_ cells has been documented in individuals with controlled viremia [125], and T_FH_ cells have been proposed as the site of HIV long-term persistency [126,127]. HIV-infected T_FH_ cells are phenotypically and functionally different than their HIV-negative counterpart, which supports a dysfunctional GCR [126,127]. Dysfunctional GCR is also supported by a decrease in CD27^pos^ memory B cells [128,129], a remarkable observation considering that memory cells are the site of EBV-unharmful persistent infection [3].

T cells with an exhausted phenotype accumulate in long-term HIV carriers [130] (Figure 2). Exhaustion is executed by a series of immune checkpoint inhibitors (ICI), and the most important ones of these are PD-1, CTLA-4, LAG3, TIGIT, and TIM3 [131]. The expression of ICI should signal for the control of acute antigenic challenges to prevent lingering responses and auto-immunity. However, in chronic antigenic stimulation, such as persistent infections and cancer, T cells expressing ICI markers accumulate, indicating untimely exhaustion [131]. Terminally exhausted T cells do not respond to antigenic stimulation, thus failing to eliminate infected/tumor cells. There are also senescent T cells that are characterized by low to no expression of costimulatory receptors CD28 and CD27 and by the lack of proliferation in response to new antigenic challenges despite an effector memory phenotype. Senescent T cells progressively lose their telomeres and exhibit a metabolic switch that is characterized by enhanced glycolysis and low mitochondrial respiration, but they do not express multiple ICI [132]. Senescent T cells are also characterized by a reduced TCR repertoire, but with enhanced specificity for CMV and EBV antigens. It has also been proposed that senescence is responsible for the lack of response to new antigens in the elderly, for instance, to influenza vaccination [133]. T cells from PLWH also exhibit reduced responses to influenza or tetanus vaccination and attrition of the TCR repertoires [134,135]. EBV viremia correlates with CD4 and CD8 T cell-reduced TCR repertoires, and CD8 T cells isolated from PLWH are not activated by lysates of lymphoblastoid cell lines [136]. Remarkably, exhausted T cells located in the lymph nodes seem to be at least one of the main reservoirs of HIV-persistent infection, probably because of the restraint that ICI expression imposes on T-cell activation [137,138], and CMV and EBV-specific CD4 T cells have been documented as a preferred and even exclusive reservoir for persistent HIV infection [139].

There are multiple levels of evidence supporting an HIV-EBV interaction in addition to the appearance of EBV-positive lymphomas. For instance, Kenyan children on ART exhibited a correlation between EBV primary infection and HIV viremia, which supports that EBV influences HIV reactivation [140]. Similarly, detectable EBV load at the initiation of ART or post-ART was associated with positive markers of inflammation, non-AIDS defined malignancies, and other PLWH morbidities, such as cardiovascular disease [141]. EBV load in the peripheral blood correlates with hypergammaglobulinemia and with the presence of LMP1-transcripts in cells, despite ART-suppressive HIV viremia and increased CD4 numbers [142]. HIV viremia also correlates with other systemic findings, such as enhanced EBV load in saliva, semen, and breast milk. A study revealed that EBV load remains detectable despite successful ART, while the CMV levels became undetectable [143]. EBV load also progressively increases with longer periods since HIV diagnosis and with HIV viremia [144,145]. EBV detection in the semen has also been linked to an increased risk of HIV sexual transmission [146,147], and EBV and HIV have been co-detected in the breast milk of the patients [148]. EBV viremia has been correlated with increased frequencies of CCR5^pos^ CD4 T cells that exhibit an enhanced susceptibility for HIV infection in vitro [149]. HIV infection rendering CD4 and CD8 T cells susceptible for EBV infection in vitro has also been documented. Here, EBV infection of HIV-positive T cells resulted in enhanced HIV replication and enhanced syncytia formation [150]. EBV primary infection and reactivation in children with HIV infection have also been associated with increased apoptosis of CD8 T cells [151]. Altogether, these studies support a close and bidirectional EBV and HIV-interacting scenario, which is beyond the archetypal immunosuppression because of the lack of T cells. They also support that the control of EBV would help control the HIV cellular reservoirs, positively impacting the PLWH general well-being, most likely antagonizing the development of not only EBV-positive but also EBV-negative lymphomas.

## 6. *Plasmodium falciparum* Resembles HIV in Its Capacity to Poly-Clonally Activate B Cells and Trigger GC Continuous Re-Entry

BL that develops in *P. falciparum* holoendemic geographic regions (i.e., sub-Saharan Africa and Papua New Guinea) is known as endemic (eBL), and it is close to 100% associated with EBV. Five different species of *Plasmodium* infect humans, of which *P. falciparum* causes the most severe forms of malaria and is the only species associated with BL. The tropical regions in America and Asia wherein malaria is mainly caused by *P. vivax* have a low incidence of BL [152]. Several mechanisms have been proposed to explain the association between EBV and *P. falciparum*, some of which resemble the EBV-HIV interactions that enhance the risk of lymphoma. For instance, enhanced B cell activation with hypergammaglobulinemia and the continuous re-entry of B cells to the GCR has also been observed [153,154]. Indeed, BL cells exhibit a phenotype similar to GC centroblasts. Furthermore, *P. falciparum* infection correlates with the enhanced expression of the activation-induced cytidine deaminase (AID) [155,156], the GCR enzyme in charge of the BCR hypersomatic mutagenesis and isotype switch recombination. The analysis of tonsils derived from *P. falciparum*-infected individuals has shown enhanced GC formation and enhanced EBV infection of GC cells. Moreover, isolated B cells turned on AID expression upon stimulation with lysates derived from *P. falciparum*-infected erythrocytes [156]. Infection with murine *P. chabaudi* also revealed enhanced GCR and AID expression [157]. GCR entry and AID expression may be responsible for the *myc* translocation that characterizes all types of BL [158]. *P. falciparum* also encodes the erythrocyte membrane protein 1 (PfEMP1), a super-antigen that poly-clonally activates B cells and protects them from apoptosis [159,160,161]. Thus, PfEMP1 also contributes to B cell hyperactivation and re-entry into GCR, similar to the heterologous B cell activation observed with HIV p17 and gp120 (Figure 2). Interestingly, PfEMP1 is not encoded by other species of *Plasmodium* [162], which potentially explains the geographical restriction of eBL.

The strict association with *P. falciparum* but not with other *Plasmodium* species remains puzzling. The severity of disease presentation does not seem the explanation, since individuals with the hemoglobin variant (HbSS homozygous or HbSA heterozygous) responsible for sickle cell anemia are protected against severe malaria but not against eBL [163]. Because of the similarities with HIV lymphomagenesis, it is unclear why *P. falciparum* and EBV coinfection only associates with eBL, but not with other types of lymphoma. Moreover, there is a lag between malaria and eBL presentation; while most children are already protected from the former at the age of 5 years, eBL often picks up later, depending on the African country [164,165]. Long-lasting parasitemia subsisting at the sub-clinical levels for several years has been proposed/observed [166]. In this regard, a recent study observed that adult patients with sporadic BL from the United Kingdom also had a history of exposure to *P. falciparum* [167].

A state of relative immunosuppression has been recorded in patients with acute malaria, with decreased cytotoxic activity and decreased IFNγ secretion in response to latent and lytic EBV antigens. T-cell dysfunction correlates with enhanced EBV reactivation/viremia and is heightened after repeated episodes of *P. falciparum* infection [168,169,170,171,172,173]. Moreover, enhanced frequencies of *exhausted* PD-1^pos^ CD8 T cells and regulatory T cells have been recorded in eBL patients who relapsed compared with long-term survivors or healthy controls [174]. Defective NK cell activity has also been observed among Kenyan children with eBL, as characterized by the lack of expression of NKp46, NKp30, CD160, and TNFα markers of NK cell competence [175]. These studies demonstrated unpaired responses to antigens from both infectious agents and increased frequencies of defective NK cells correlated with enhanced EBV viremia. It thus seems to be interesting to compare the phenotypic and functional similarities between the T cell and NK cell defective populations induced by persistent *P. falciparum* infection with the T-cell HIV-induced exhaustion or senescence phenotypes.

## 7. EBV Interactions with Oral Bacteria May Facilitate Viral Transmission and Promote Periodontal Diseases

EBV-associated lesions often arise at places that have been the site of residence or colonized by other microbes. Although the presence of EBV together with these agents often promotes or aggravates the disease, it is unclear whether they involve some type of direct or indirect interactions. The oropharynx, as the site of continuous release of infective EBV particles, provides plenty of opportunities for EBV to cross paths with resident or transient pathogens. Particularly, the codetection of EBV and oral bacteria have been associated with periodontal disease. EBV has been detected in deep periodontal pockets in great numbers of up to 2.8 × 10^9^ virus/mL [176], and, in cases of refractory diseases, also in the numbers of almost a million copies/mL [177]. Meta-analyses of 50 studies have revealed that while EBV was detected in only 8% of healthy periodontium samples, in chronic and aggressive periodontitis, it increased to 39% and 52%, respectively [178,179]. Miller et al. treated EBV-positive periodontal patients with valacyclovir and noted a reduction of EBV PCR-positive samples from 29 to 18 after only 3 days of treatment [180], while Sunde et al. observed a 60% reduction in the viral load in the periodontal plaque of a patient treated with the same antiviral agent [177]. The fact that valacyclovir was administered in the absence of antibiotics strongly supports that the sole reduction of EBV helps improve the clinical status of the lesion. EBV has been reported to directly infect epithelial cells from diseased gingival tissues via in situ hybridization (ISH) studies, while healthy tissues were negative for EBV [181]. The viral load in periodontal tissues has been correlated with disease severity, and EBV positivity precedes the onset of the disease [182].

Periodontitis is a poli-microbial disease triggered by dysbiosis of the oral biota. Although the mouth harbors distinct habitats comprising one of the most diverse microbial communities of the human body, *Aggregatibacter actinomycetemcomitans* and *Porphyromonas gingivalis* are key low-abundance periodontal bacteria that, when over-grown, can bring these ecosystems to a dysbiotic state, particularly the gingiva [183]. An association between EBV and *P. gingivalis* in juvenile periodontitis has also been suggested [184]. In addition, the odds ratio (OR) of finding *A. actinomycetemcomitans* in localized juvenile periodontitis increased nine-fold when EBV was co-detected [184]. Another study on juvenile periodontitis reported ORs of 49 and 10.7 of EBV and *A. actinomycetemcomitans* or *P. gingivalis* coinfection, respectively [185]. Multiple studies conducted on patients with adult periodontitis around the world have also observed EBV association with *P. gingivalis* and *A. actinomycetemcomitans* [186].

Considering the large numbers of EBV copies reported, and the steady shedding of EBV in the saliva [187], EBV interactions with the oral microbiota may represent a mechanism that evolved to facilitate EBV local reactivation and viral transmission to new hosts. Indeed, *P. gingivalis* and other resident bacteria produce butyric acid, a short-chain fatty acid that is a metabolic end-product of obligate anaerobes. Butyric acid is a potent class I histone deacetylase (HDAC) inhibitor, and the EBV lytic cycle is importantly controlled by HDACs [188,189]. Butyric acid is widely used to induce the EBV lytic cycle under experimental settings. *P. gingivalis*, *Fusobacterium nucleatum*, and *Porphyromonas endodontalis* reactivate EBV-infected B cells through induction of the *BZLF1* expression [190,191]. Moreover, chromatin immunoprecipitation assays have revealed the dissociation of HDAC1, 2, and 7 from the *BZLF1* promoter after EBV-infected B cells were treated with conditioned media derived from the cultures of the periodontopathic bacteria [191]. Moreover, high concentrations of butyric acid are present in the periodontal pockets and saliva [192,193]. Butyric acid is also a known potent activator of the HIV lytic cycle [194]. Here, however, contrary to the oral niches, GI mucosal dysfunction has been associated with decreased levels of butyric acid because of the microbiota change from butyrate-producing Firmicutes to Bacteroidetes [195]. In this scenario, intestinal low levels of butyric acid may favor the EBV and HIV latency stages, rather than the reactivation stages, perhaps impacting the synergic lymphomagenic mechanisms of these viruses. It is also worth mentioning that other herpesviruses have also been implicated in the etiology of periodontitis. For instance, mice coinfected with *P. gingivalis* and CMV exhibited higher mortality rates and lower levels of IFNγ than mice infected only with the bacterium, which supports cooperation in the pathogenic mechanisms [196]. Oral transmission is the most common mechanism for the dissemination of most human herpesviruses. *A. actinomycetemcomitans* has also been reported to reactivate EBV through the secretion of the bacterial toxin cytolethal distending [197].

## 8. HPV Is Also a Tumor Virus That Inhabits Oral and Genital Tissues

High-risk HPV16 and HPV18 are the main causes of cervical squamous cell carcinomas (CESC), accounting for nearly 100% of all CESC cases. It has been estimated that only 3.6% of cervical low-grade squamous intra-epithelial lesions (LSIL) progress to a high-grade lesion (HSIL), and approximately half of the progressing lesions eventually regress despite carrying high-risk HPVs [198], indicating additional cooperating factors. EBV has been proposed as one of the cofactors that aggravate HPV infection. The presence of EBV in cervical lesions has been addressed by ISH and immunohistochemistry/immunofluorescence of viral proteins, with some studies supporting a progressive increase in positivity from normal to LSIL to HSIL/CESC [199]. These detection methods have confirmed that EBV positivity arises from the epithelial cells rather than from infiltrating B cells. Some of these data are quite compelling, reporting a positivity of 0–16% in normal or LSIL to 40–80% in advanced lesions [200,201,202,203,204]. Overall, these studies support that EBV increases the risk to develop CESC by almost 4-fold. EBV-HPV coinfection has also been associated with more aggressive tumors [205]. On the contrary, other studies have failed to detect any EBV infection on cervical epithelial cells or it has been detected irrespective of the progressing state of the lesion [206,207,208,209,210,211]. Moreover, arguing that EBV only plays a passenger role in cervical lesions, infection is only present in a small subset of tumor cells, with an estimated load of one viral copy per tumor cell [203]. The reason for these discrepancies is presently unclear. The recent review by Blanco et al. dissects all the studies addressing the EBV and HPV copresence in the cervical tissues and the methods used to determine EBV infection [212].

The aforementioned studies could not conclude the possible EBV and HPV co-influence nor the cooperative mechanisms displayed by both viruses. More mechanistic approaches have demonstrated a decreased EBV-specific cytotoxic immune response in advanced lesions when compared with that in early lesions, relating to EBV-induced immunosuppressive mechanisms [213] (Figure 3). The EBV EBNA2 expression has been detected in advanced coinfected samples, which is suggestive of a viral latency III program and also of an immunosuppressive environment [203,204]. In addition, the more active the cytotoxic activity against EBV, the better the survival chance of the CESC patient, regardless of the status of HPV infection [213]. A different line of evidence associates the presence of EBV with a 5–7-fold increased probability of HPV18 and HPV16 integration, respectively [214,215]. Moreover, increased methylation of tumor suppressor gene promoters has been indicated in EBV-HPV coinfected specimens [216]. Unfortunately, in these studies [214,215,216], the detection of EBV was performed by PCR, which did not distinguish it from epithelial or B cell infection.

HPV has also been associated with approximately 35% of the head and neck squamous cell carcinomas (HNSC) [217]. Nasopharyngeal carcinoma (NPC) is a subtype of HNSC that has been associated with about 100% of EBV infection, particularly the non-keratinizing subtype. HPV has been codetected in 10–47% of NPC from the endemic regions, which are EBV-positive cases [218,219,220,221]. On the contrary, in the study by Lin et al., no HPV-positive NPC was detected in an endemic region in China [222]. In agreement, NPC from the non-endemic regions was either EBV or HPV positive, but not double-positive [222,223,224,225,226]. Another ISH-based study supported EBV-HPV coinfection in 25% and 20% of tonsillar and base of the tongue oral squamous tumors, respectively [227]. High-risk HPV16 and HPV18 have been reported as the most common HPVs, which are more often associated with EBV-coinfection in HNSC.

A common requirement for HPV and EBV persistency in epithelial cells is an undifferentiated state, switching to lytic replication once the cell assumes a differentiation pathway [25,228,229]. EBV LMP2A delays the nasopharyngeal epithelial cell differentiation [230], and HPV oncogenes E6 and E7 do the same in tonsillar organotypic raft cultures, immortalized oral keratinocytes, and primary human foreskin keratinocytes [231,232]. A permissive environment for EBV persistence may be created by HPV-related delayed activity of KLF4 and BLIMP1 [231], two of the central transcription factors required for the EBV lytic cycle in differentiating epithelial cells and B cells [233] (Figure 3). A similar program of EBV gene expression has been reported in HPV-infected CESC and oral carcinomas, with positivity for LMP1, EBNA1, BARF1, and EBNA2 [199,234]. LMP1 and E6 double recombinant expression in murine embryonic fibroblasts associated with oncogenic events, such as enhanced NFκB signaling, increased proliferation and resistance to apoptosis in vitro, and tumorigenesis in nude mice [235] (Figure 3). EBV and HPV coinfected primary oral keratinocytes also demonstrated an increased degree of invasiveness [227].

## 9. *Helicobacter pylori (Hp)*, Inflammation, and EBV in the Transformation of Gastric Epithelial Cells

Among the organs in which EBV intersects with other pathogens, perhaps one of the most surprising ones is the stomach. Unlike the oropharynx, the stomach lacks secondary lymphoid tissues to support viral latency and may seem like a dead-end for EBV-productive replication. Nevertheless, EBV is responsible for nearly 10% of all gastric cancers (the EBVaGC), and the EBVaGC outnumbers any other EBV-associated cancer. Gastric cancer is a neoplasm with a clear inflammatory etiology. Hp, a stomach resident bacterium, has been considered as the main trigger factor of the inflammatory process.

Direct and indirect EBV-Hp interactions have been documented (see below). However, Hp has been associated mainly with cancers of the distal portions of the stomach region, while EBV has been more associated with cancers of the proximal regions [236], suggesting that both pathogens do not often coexist at the stomach locoregional level. However, an ongoing inflammatory process, most likely induced by Hp, may be responsible for the chemoattraction and reactivation of EBV-infected B cells, which explains the localization of EBV in the gastric mucosa. Evidence of inflammation favoring EBVaGC has been proposed from the tumors arising at the surgical site of partial gastrectomy or gastroenterostomy, the so-called stump or remnant gastric carcinomas, which are up to 35% EBV-positive [236]. Similar to the EBV and HPV interactions, Hp-induced mutagenesis may also shape the cellular genetic background that licenses EBV persistence in the epithelial cells.

Several studies have assessed the presence of both pathogens in gastric lesions [237]. PCR-based studies have observed a significant association between coinfection and more advanced inflammatory lesions [238,239,240,241,242] and between Hp infection and increased EBV load-supporting Hp-induced viral reactivation [240,242]. ISH has often failed to confirm such an association in the pre-neoplastic lesions [243]. Hp-positive EBVaGC exhibits an increased risk for distant metastasis compared with only EBV-infected neoplasms [244]. We reported increased anti-EBV antibody titers correlating with increased anti-Hp titers and with severe inflammation in the pre-neoplastic and neoplastic lesions [245,246]. We made the same observations in pediatric samples, wherein only children seropositive to both agents were associated with severe gastritis [247].

Monochloromine is a membrane-permeating oxidant produced by neutrophils in response to Hp gastric colonization. In vitro experiments have demonstrated that monochloramine reactivates EBV from latently infected B cells and epithelial cell lines [248]. In another in vitro model, CagA could increase EBV permissive infection of a gastric epithelial cell line, although the infection seemed lytic rather than latent [249]. CagA is a Hp virulence factor and a bacterial oncogene that is transduced into the epithelial cells, perturbing important oncogenic signaling pathways [250]. The CagA signaling activity is importantly driven by phosphorylation, and, Saju et al. reported that SHP1 phosphatase counteracted CagA activity [251]. EBV epigenetically silenced the *SHP1* promoter enhancing the pro-oncogenic activity of CagA. Moreover, *SHP1* promoter hypermethylation and *SHP1* downregulation have been confirmed in EBVaGC. EBV and Hp coinfected cells have been reported to show an enhanced expression of DNA methyl transferases (DNMT) and hypermethylation of tumor suppressor gene promoters, leading to a decreased expression of key regulatory enzymes participating in the cell cycle, apoptosis, and DNA damage repair [249] (Figure 3). CagA can also be released in bacterial-like exosomes, widening the area of CagA influence [252]. To date, most studies addressing a potential synergy between EBV and Hp have been restricted to EBVaGC, mostly exploring how Hp influences EBV. Whether EBV is also an important trigger of inflammation and gastric mucosa damage has only been indirectly explored [245,246].

## 10. Conclusions

Although most people live as EBV carriers without showing any clinical manifestation, EBV has the potential to trigger life-threatening conditions. This situation is indicative of how well EBV has adapted to humans as well as illustrative of its pathobiont capacity. Beyond immunosuppression, it remains unclear why healthy people also develop disorders associated with this virus. Increasing lines of evidence support that EBV crosses paths with HIV, *P. falciparum*, and KSHV pathogenic mechanisms, which enhances the risk of developing lymphomas. These agents directly or indirectly stimulate B cells, which in turn activates the GCR, a highly mutagenic process that increases the risk of lymphomagenesis. Lymphomagenesis may also be aided by mechanisms undermining cancer immunosurveillance, such as T cell exhaustion and senescence. Bacterial products may attract EBV-infected B cells and promote its reactivation in tissues, such as butyric acid by oral bacteria and monochloramine by Hp. This mechanism may have evolved in the oral cavity to facilitate the transmission of EBV to new hosts. HPV and KSHV share the feature of effective modulation of epithelial and B cell terminal differentiation. It is believed that HPV- and KSHV-induced cell arrest in less mature stages may provide more suitable conditions for EBV latent infection. Likewise, Hp-induced mutagenesis may provide the cellular genetic background for EBV persistence in gastric cells.

Several other interactions between EBV and infectious agents have been proposed, although studies that rely only on the copresence of different agents do not necessarily explain the etiopathogenesis of the resultant diseases. Furthermore, these observations are often made through PCR-based techniques, which do not inform about the nature of an EBV-infected cell or the number of infected cells. EBV persists in B cells, and these cells continuously patrol the tissues and organs; hence, positivity does not indubitably mean causality. Even in studies in which the presence of EBV appears to worsen the disease, it is not easy to distinguish whether EBV is causal or an innocent bystander. Gastric cancer exemplifies this ambiguity. Although EBV and Hp cause gastric cancer, it remains unclear whether the emergence of this neoplasm is potentiated after some type of interaction between them or whether both agents transform the gastric epithelial cell through independent mechanisms. Similarly, in transplanted patients, the codetection of EBV and betaherpesviruses has often been correlated with enhanced tissue damage, graft rejection, and PTLD. However, there is still a long way to go before establishing synergies in the infectious cycles of these viruses. The more thoroughly we document the intersections of EBV with other pathogens, the better we would be able to understand the circumstances that lead to its pathogenicity. Unveiling the mechanism(s) responsible for uncontrolled EBV infection is expected to provide the points of prophylactic or therapeutic interventions.

## Figures and Tables

**Figure 1 viruses-13-01399-f001:**
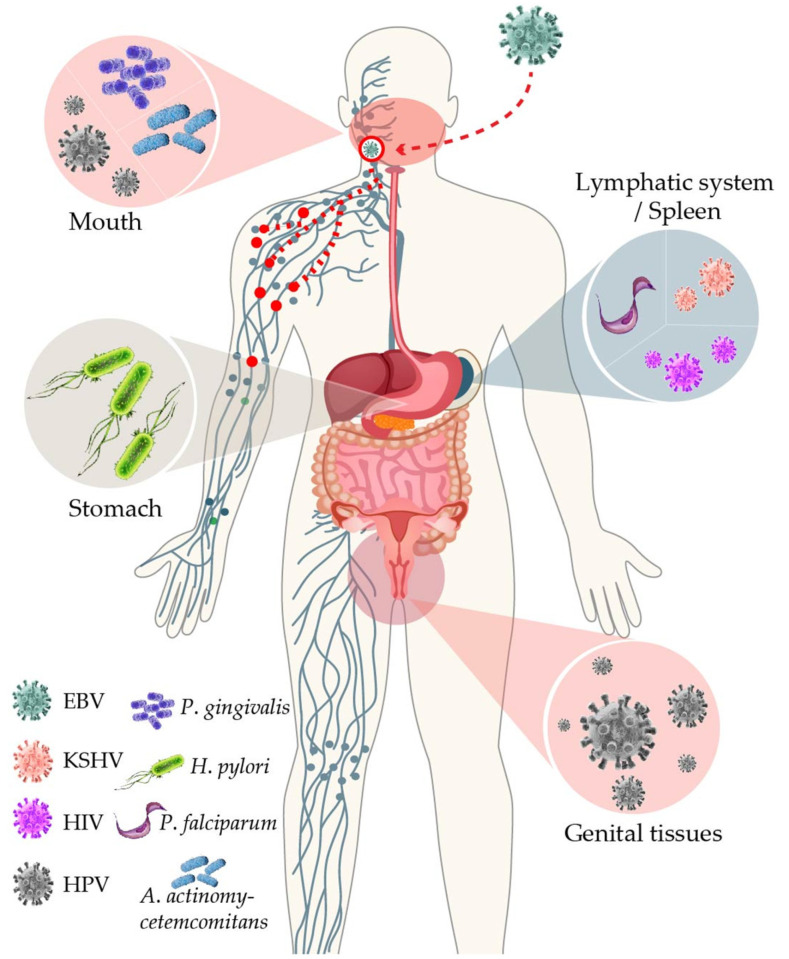
Shared microenvironments between EBV and other infectious agents. EBV infection’s main portal of entry and exit is the oral mucosa, where it coexists with the periodontal bacteria *P. gingivalis* and *A. actinomycetemcomitans*, and HPV. EBV persists in B cells residing in the lymphatic system, where it can cross paths with other infectious agents such as HIV, *P. falciparum*, or KSHV (here represented in the spleen as an example of a secondary lymph node). From the lymphatic system, EBV-infected cells can infiltrate certain tissues, such as the stomach or cervix, where EBV can intersect with *H. pylori* and HPV, respectively.

**Figure 2 viruses-13-01399-f002:**
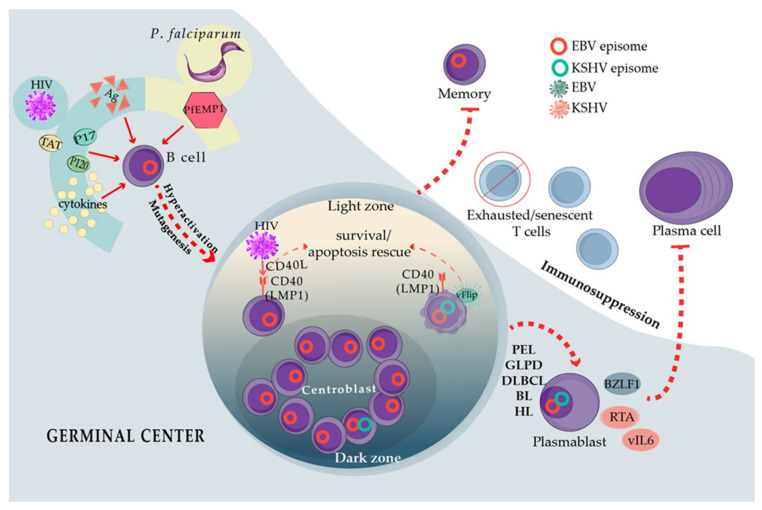
EBV-infected B cells transiting through the germinal center reaction (GCR) cross paths with HIV, *P. falciparum*, and KSHV. The key convergent mechanisms that potentiate lymphomagenesis are antigen (Ag)-specific and heterologous **hyperactivation** of B cells, which, upon enhancing the passage through the GCR, increase the likelihood of off-target **mutagenesis**. Mutated B cells normally die of apoptosis. However, if EBV alone or in conjunction with KSHV resides in these lymphocytes, viral proteins can rescue the cells from dying (illustrated here with LMP1 and vFLIP). In addition, the transformed/mutated B cells should be eliminated by the immune system, but infection-induced **exhaustion** or **senescence** causes unpaired immunosurveillance and immunoescape of these cells. The viral products also activate the oncogenic pathways and oncogenic processes and block B cell differentiation into memory cells. Immunosuppression by viral orthologs of cellular cytokines may also contribute to lymphomagenesis. Altogether, these events result in the development of conditions that favor the emergence of lymphomas with a plasmablastic morphology (see text for a more detailed explanation): Burkitt’s lymphoma (BL), Hodgkin’s lymphoma (HL), diffuse large B cell lymphoma (DLBCL), primary effusion lymphoma (PEL), and germinotropic lymphoproliferative disorder (GLPD).

**Figure 3 viruses-13-01399-f003:**
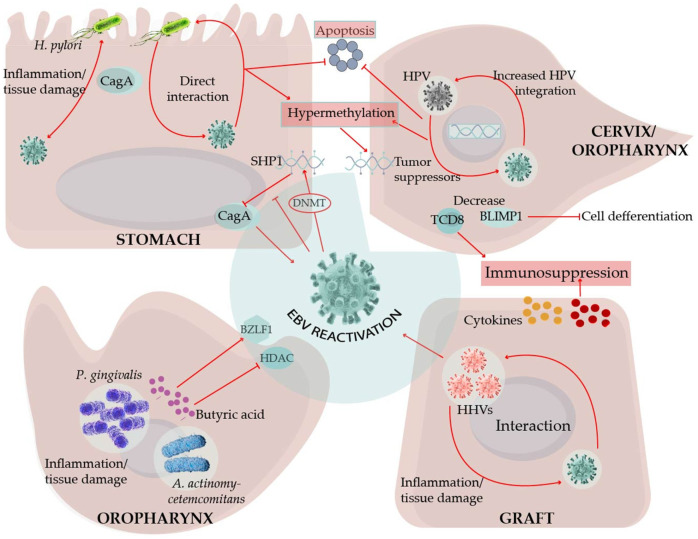
Interactions between EBV and other infectious agents in non-lymphatic organs. The arrival and reactivation of EBV-infected B cells in different tissues may be influenced by ongoing inflammation. Reactivation is better understood by the secretion of butyric acid by oral bacteria. Released viral particles can infect epithelial cells aggravating inflammation and worsening tissue damage. EBV together with *H. pylori* or HPV alter the expression of methyltransferases (DNMT), which silences the tumor suppressor genes, such as the phosphatase SHP1, rendering the bacterium oncoprotein CagA more active. *H. pylori*, HPV, and EBV products also enhance the survival of cells. EBV and HPV also turned off the expression of differentiation genes, such as *BLIMP1*, thus halting cells in the proliferative states to become more permissive for viral persistency, and EBV may favor HPV integration into the host DNA. EBV immunomodulatory mechanisms may help create an immunosuppressive microenvironment that may cooperate with the immunomodulatory mechanisms of other herpesviruses (HHV). Altogether, these interactions promote the appearance of tissue lesions that may be partially responsible for graft rejection and carcinogenesis (see text for details).

## Data Availability

Not applicable.

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
