# Peer review of "The Role of Coinfections in the EBV–Host Broken Equilibrium"

_viruses, 2021, doi:10.3390/v13071399_

Round 1

Reviewer 1 Report

This manuscript is a comprehensive review of the literature on how different viral/bacterial infections may co-operate to exacerbate EBV-associated disease. It contains some interesting potential links between the infectious agents which is good to have all in one document. However, the manuscript needs considerable work.

Major comments.

The authors use non-scientific language throughout the manuscript, some of which is inappropriate. For example, line 50; helping to shape the mechanism of truce and line 62; EBV quickly ‘finds’ and infects B cells. These terms are appropriate as they anthropomorphise the virus.

Line 53/54. EBV is a true pathogen, not just part of our normal biota. It can cause disease following primary infection, including IM. The authors present no evidence to the contrary.

The authors need to be more accurate with statements. For example: Paragraph beginning Line 70. There is no evidence that EBV oscillates between latencies once latency is established in B cells. The authors need to state what we know happens, ie EBV infects B cells, initiates latency III. Lat III is thought to be downregulated during the GC reaction where the infected memory B cells emerge as lat 0, with just expression of the EBERs and miRNAs. Thus, the latency profiles change due to specific circumstances, ie transit through the GC following primary infection; differentiation into plasma cells.

Another example of where statements need to be improved; EBV in the immunocompromised individual, line 89. This section needs to be more accurate. The authors need to state how the virus is controlled by the various components of the host immune response, ie what exactly is the immune response to the virus during primary infection then what happens when individuals are immunocompromised. At the moment, it is unclear.

It is not clear what the title of section 3 means.

The authors need to focus on the subject of each section and cut out irrelevant material. For example; when talking about PEL, we do not need to know about mutations/translocations that are present in other lymphomas, especially when you are addressing these lymphomas in later sections. Also, this section, like many other sections, needs to be more focussed. At the moment, it is confused. The authors bring in subjects without having explained them properly. For example; line 154, ‘KSHV LANA switches off the Cp promoter’, but do not really explain what Cp is and in which virus. The authors need to carefully explain each concept logically.

Section 4. Co-infection of EBV and betaherpesviruses. The authors need to be very clear in this section what they are trying to say. Given the majority of individuals are infected with EBV, most individuals infected with the betaherpesviruses will also be co-infected with EBV. So, the most important concept here would surely be reactivation of the viruses? How often is EBV plus another herpesvirus reactivation noted in transplant patients? If both viruses present, does viral load matter? Or is it only important when the virus loads reach a threshold level?

The same needs to be addressed for CMV and EBV in PTLD. What is the important question in this section? EBV obviously causes a subset of PTLD, so how does CMV influence the occurrence of PTLD?

Acyclovir is not used to treat either EBV or CMV (line 252).

Line 263. EBV and HHV6 do not infect the same cell type.

Although it is nice to think that EBV and HPV may co-operate in cervical cancer and in head and neck cancers, there is currently very little evidence of HPV and EBV co-infection in the same cell. Furthermore, there is actually very little evidence that EBV is transmitted sexually. This section needs to be very balanced if it is going to be included. The majority of studies do not find a connection between the viruses. It is important to remember that EBV does not establish a latent infection in normal epithelial cells; only lytic reactivation is observed. This is evident in NPC with several mutations essential before latency can be established. Therefore, the authors need to address how the two viruses will co-operate when it is likely that HPV will probably need to do the damage before EBV latency can be established.

The final section on helicobacter and EBV needs to be far more focussed as it contains a lot of irrelevant material. The authors need to look at stages of gastric disease where EBV is actually found in the epithelial cells. Is HP present at this stage of disease? Does HP cause inflammation that brings EBV in B cells? Evidence?

Author Response

REVIEWER NO. 1

Comments and Suggestions for Authors

This manuscript is a comprehensive review of the literature on how different viral/bacterial infections may co-operate to exacerbate EBV-associated disease. It contains some interesting potential links between the infectious agents which is good to have all in one document. However, the manuscript needs considerable work.

  1. Thank you very much for your time and consideration to review the manuscript. We would like to emphasize that your comments were thoroughly considered. We significantly reduced the length of the review, so as not to give the impression that it is out of focus or that the studies included were not carefully inspected. The word count was taken from 10,001 to 8597 in the modified manuscript. The manuscript was also sent to English style correction (see uploaded certificate).

Major comments.

  1. The authors use non-scientific language throughout the manuscript, some of which is inappropriate. For example, line 50; helping to shape the mechanism of truce and line 62; EBV quickly ‘finds’ and infects B cells. These terms are appropriate as they anthropomorphise the virus.
  2. Although some informal language was chosen to help the reading of the review, the terms that anthropomorphise the virus were eliminated in the modified manuscript.
  3. Line 53/54. EBV is a true pathogen, not just part of our normal biota. It can cause disease following primary infection, including IM. The authors present no evidence to the contrary.

We support that EBV is more a pathobiont than a true pathogen based on that close to 100% of the adult human population is infected with EBV and only a relatively small proportion develops disease. Most humans are carriers of the virus since childhood, and we live our entire lives without overt manifestation of disease. We believe that it is important to say it this way because the review is based on the idea that an important trigger of a dysbiotyc condition is when EBV cross paths with other pathogens. IM is probably also a form of dysbiosis since not all infected individuals develop IM. IM is actually very rare in developing countries. IM is only tangentially mentioned in the review because the reports that CMV and EBV codetection is associated with IM.

  1. The authors need to be more accurate with statements. For example: Paragraph beginning Line 70. There is no evidence that EBV oscillates between latencies once latency is established in B cells. The authors need to state what we know happens, ie EBV infects B cells, initiates latency III. Lat III is thought to be downregulated during the GC reaction where the infected memory B cells emerge as lat 0, with just expression of the EBERs and miRNAs. Thus, the latency profiles change due to specific circumstances, ie transit through the GC following primary infection; differentiation into plasma cells.

We have been more careful with our statements throughout the manuscript. We think that EBV probably oscillates between latencies since it resides in memory B cells, and this is a very dynamic population. Memory B cells can differentiate into a plasma cell or re-enter the germinal center reaction with the appropriate signals, while the former would send EBV into the lytic cycle, the virus most probably adjusts to the latter exiting latency 0. The reviewer is right in that we do not know what exactly happens with the EBV latent transcriptional programs when a memory B cell re-enters the germinal center reaction and exits again as a memory B cell.

  1. Another example of where statements need to be improved; EBV in the immunocompromised individual, line 89. This section needs to be more accurate. The authors need to state how the virus is controlled by the various components of the host immune response, ie what exactly is the immune response to the virus during primary infection then what happens when individuals are immunocompromised. At the moment, it is unclear.
  2. The reviewer is right in that we still lack knowledge about how EBV is controlled by the immune system. We have modified this sentence including a sentence stating that. We have also added a reference of a review of the immune response to EBV (reference #4). We mention the immunocompromised individual only as one of the clearer conditions of dysbiosis that exacerbates EBV infection/reactivation, but it is beyond the aim of this review to explain the immune response that controls EBV.  
  3. It is not clear what the title of section 3 means.

  1. According to the reviewer observation, we have simplified this and other titles in the manuscript to facilitate reading. The title of section 3 now reads "EBV and KSHV coinfect B cells in primary effusion lymphoma and germinotropic LPD". Hoping it is clearer now.
  2. The authors need to focus on the subject of each section and cut out irrelevant material. For example, when talking about PEL, we do not need to know about mutations/translocations that are present in other lymphomas, especially when you are addressing these lymphomas in later sections. Also, this section, like many other sections, needs to be more focused. At the moment, it is confused. The authors bring in subjects without having explained them properly. For example, line 154, ‘KSHV LANA switches off the Cp promoter’, but do not really explain what Cp is and in which virus. The authors need to carefully explain each concept logically.
  3. Following the reviewer recommendation, we restructured the entire KSHV section, eliminated the least relevant information, and explained better each concept throughout all the manuscript.
  4. Section 4. Co-infection of EBV and betaherpesviruses. The authors need to be very clear in this section what they are trying to say. Given the majority of individuals are infected with EBV, most individuals infected with the betaherpesviruses will also be co-infected with EBV. So, the most important concept here would surely be reactivation of the viruses? How often is EBV plus another herpesvirus reactivation noted in transplant patients? If both viruses present, does viral load matter? Or is it only important when the virus loads reach a threshold level?
  5. Thank you very much for the comment. In the revised version of the manuscript, we tried to explain better that although these viruses are ubiquitously present in the population, they are generally undetectable or detectable with low viral loads in immunocompetent carriers. We also mention that the frequency of codetection of EBV and other herpesviruses ranges from 2.6% to 32.7% in the post-transplanted patient (line 240-241 of the document without Track changes). We have added sentences disclosing the information regarding viral load detection thresholds, using EBV as an example, which is the one for which there is more information and the object of this review. We also tried to convey that there are no clear-cut numbers in the literature and why, and that the proposed thresholds are often arbitrary and heterogeneous. For instance, most studies support that CMV and EBV codetection is a determining factor for graft dysfunction and graft rejection independently of viral loads. On the contrary, Barani et al. found a greater number of recipients presenting graft dysfunction with "low" viral loads (1000-2000 for EBV) than with "high" viral loads (>5000 copies/ml of whole blood). According to Barani et al, some graft recipients have "high" viral loads and remain stable for months or years without being associated with rejection or developing PTLD (Reference #55). We also mention in the revised manuscript that the proposed viral loads for EBV range from 500 to 4000. Finally, we were more careful using the term infection and replace it for detection or reactivation.

The same needs to be addressed for CMV and EBV in PTLD. What is the important question in this section? EBV obviously causes a subset of PTLD, so how does CMV influence the occurrence of PTLD?

  1. There are no studies addressing the specific mechanisms by which these herpesviruses cooperate to trigger/enhance the risk of PTLD. We found one example in the literature in which the codetection between EBV and CMV was associated with PTLD in immunocompromised children. These children also exhibited a decreased frequency of CD56dim NKG2Apos KIRneg NK cells, which was associated with loss of control of EBV (Reference #59). We also mention that prophylactic treatment with ganciclovir decreases the rate of PTLD. Therefore, as mentioned throughout the text, most studies are only associative.

Acyclovir is not used to treat either EBV or CMV (line 252).

  1. We are not sure what the reviewer means here, acyclovir and other drug members of the acyclovir family are usually used as prophylactic treatment to herpesvirus infections. These drugs target the thymidine kinase present in all herpesviruses. This is an enzyme expressed during the viral lytic cycle, and EBV associated diseases linked to latency are not responsive to treatment. For instance, for IM, although it does not solve IM clinical symptoms it does reduce EBV shedding in saliva. We have been more careful to specify the specific drug used in the studies mentioned, changing to ganciclovir and valgancyclovir.

Line 263. EBV and HHV6 do not infect the same cell type.

  1. In the referred study, PBMCs were infected with each virus individually or together. The study does not address whether the virus co-infect the same cells but observed a switch in the level of cytokines (lower levels) after infection with both viruses. The study suggests an indirect mechanism to influence cytokine secretion.

Although it is nice to think that EBV and HPV may co-operate in cervical cancer and in head and neck cancers, there is currently very little evidence of HPV and EBV co-infection in the same cell. Furthermore, there is actually very little evidence that EBV is transmitted sexually. This section needs to be very balanced if it is going to be included. The majority of studies do not find a connection between the viruses. It is important to remember that EBV does not establish a latent infection in normal epithelial cells; only lytic reactivation is observed. This is evident in NPC with several mutations essential before latency can be established. Therefore, the authors need to address how the two viruses will co-operate when it is likely that HPV will probably need to do the damage before EBV latency can be established.

  1. We are sorry we gave the impression of an unfocussed unfiltered review of the studies addressing potential EBV and HPV interactions. There are hundreds of studies in PubMed addressing this question (650 articles were retrieved). For this review, we prioritized the studies in which EBV ISH (the gold standard for EBV detection) or immunofluorescence were performed, and in which infection of epithelial cells was observed. We also prioritized the studies in which the codetection of both viruses was associated with a clinical or molecular future. For instance, associated with higher viral loads, more viral integration, worse presentation of the disease, etc. When the study supports an important clinical association, but EBV was only demonstrated by PCR, we mention it as a limitation of the study. Studies using the same techniques but in which no association was found, are also mentioned in the review. We did this for all the viral associations included in the manuscript. In the revised manuscript, we eliminated “controversial” studies whose conclusions are only coming from one study. For instance, claiming that CD21 is expressed in the epithelial tissue of the cervix. We also mention how HPV interferes with epithelial cell differentiation, and how this may provide the conditions for EBV persistent infection.

The final section on helicobacter and EBV needs to be far more focussed as it contains a lot of irrelevant material. The authors need to look at stages of gastric disease where EBV is actually found in the epithelial cells. Is HP present at this stage of disease? Does HP cause inflammation that brings EBV in B cells? Evidence?

  1. We have reduced this section focusing on the evidence of the EBV and Hp interactions. The search for EBV in pre-neoplastic lesions have been conducted in two manners: 1) looking for the virus in the non-tumor tissue surrounding the tumor and 2) comparing patients with actual progressing lesions with patients without gastric lesions (e.g. non-atrophic gastritis). Although the former has given mostly negative results, the latter has found the virus preferentially or enriched in the more advanced lesions. We emphasized those studies with a statistically significant enrichment of EBV in progressing lesions. Also, since the stomach lacks lymphoid tissue and it is not a site of residency of memory B cells, we favor that is Hp, more specifically Hp-induced inflammation, what attracts EBV infected B cells to the stomach. There is little evidence for that, as mentioned in the review. However, EBV is found more often in remnant (stump) gastric cancer, probably also as a result of an ongoing inflammatory process. Monochloramine produced after Hp infection has been found to chemoattract EBV infected B cells in in vitro studies.

Reviewer 2 Report

This review article by Sánchez-Ponce and Fuentes-Pananá describes how co-infections exasperate EBV pathologies. The authors provide a detailed and thorough discussion on multiple infectious agents that contribute to EBV cancers.

Major comments:

  1. The authors need to reconsider some of the literature cited as there are 285 references, many of which are irrelevant, unnecessary, or off-topic. For example, are some of the older references really necessary (#53, #138, #122, #178 is from 1965)? Reference #21 is a two-line commentary for the study by McHugh et .al., 2017. Information from references #11, #14, #16, and #22 are likely covered in a more recent review on PEL which could be cited.

Additionally, references need to be provided for these statements:

Line 65: “B cells providing the niche for persistence” Is this referring to Thorley-Lawson’s persistence model?

Line 99: “smooth muscle cell sarcomas and carcinomas”

Line 179: “is assisted by EBV, whose transforming capacity is notably superior”

Line 362: “Receptor-less B cells should die of apoptosis…but they may be rescued by EBV latent genes” Perhaps refer to Fish, PNAS, 2020 and Mancao, Blood, 2005.

  1. Sections should stay on topic. This would also help with the massive number of references that are cited. A couple of examples:

Line 696: Reference #262, interesting but off topic.

Lines 712-713: Interesting, but off topic.

  1. The manuscript would substantially benefit from proof-reading for English language- particularly the introduction- as it is very difficult to digest in places. Words need to be corrected throughout the article to make the text more readable. Below are just a few examples:

Line 60: Via should be “route” in the sentence “EBV main via of transmission”.

Line 64: “released to the exterior”. Perhaps what is meant here is released into saliva?

Line 74 and 80: “latencies” should be latency programs.

Line 105: “bestowed severe immunosuppression” Please reconsider word choice here- bestow means “to gift or honor”.

Line 464: “EBV primo-infection” Perhaps primary infection is meant here?

Line 695: “increases de risk”

Author Response

REVIEWER NO. 2

Comments and Suggestions for Authors

This review article by Sánchez-Ponce and Fuentes-Pananá describes how co-infections exasperate EBV pathologies. The authors provide a detailed and thorough discussion on multiple infectious agents that contribute to EBV cancers.

  1. Thank you very much for your time and consideration to review the manuscript. We would like to emphasize that your comments were thoroughly considered. We significantly reduced the length of the review, so as not to give the impression that it is out of focus or that the studies included were not carefully inspected. We were surprised by the number of studies addressing EBV coinfection with other pathogens, there are literally thousands of studies only for HIV and EBV, and hundreds for each one of the other pathogens depicted in the review. Thousands of studies were not considered based on that they did not have a clinical or mechanistic implication, or because EBV was assessed by PCR. Topics in which we found a meta-analysis, this was preferred over the original articles. In the revised manuscript, we did an additional filtering of references eliminating those that did not contribute with the overall idea of the review. The references were cut from 285 to 253 in spite that we added all the references suggested by both reviewers. The manuscript was also sent to English style correction (see uploaded certificate).

Major comments:

  1. The authors need to reconsider some of the literature cited as there are 285 references, many of which are irrelevant, unnecessary, or off-topic. For example, are some of the older references really necessary (#53, #138, #122, #178 is from 1965)?
  2. We eliminated reference #53 and #178 as suggested.

Reference #21 is a two-line commentary for the study by McHugh et .al., 2017.

  1. Following your suggestion, we have left only the McHugh citation.

Information from references #11, #14, #16, and #22 are likely covered in a more recent review on PEL which could be cited.

  1. We restructured this section and eliminated some of the unnecessary references.

Additionally, references need to be provided for these statements:

Line 65: “B cells providing the niche for persistence” Is this referring to Thorley-Lawson’s persistence model?

  1. Yes, we support that memory B cells are the main reservoir of EBV persistent infection. Two references were relocated to support this statement, a Thorley-Lawson comprehensive review of the topic (reference #2) and the original paper in which EBV is found in memory B cells (reference #3).

Line 99: “smooth muscle cell sarcomas and carcinomas”

  1. This sentence was eliminated in the revised manuscript to stop adding more references and particularly old references. This also because smooth muscle cell sarcomas are less frequent since ART implementation.

Line 179: “is assisted by EBV, whose transforming capacity is notably superior”

  1. This statement is based on the capacity of EBV to transform B cells and the inability to KSHV to do so. We have supported this statement with the reference by Faure 2019 (reference #11). In the Faure study, KSHV does not infect B cells unless EBV is present. In the latter condition, 2.5% of infected PBMCs sustain both viruses.

Line 362: “Receptor-less B cells should die of apoptosis…but they may be rescued by EBV latent genes” Perhaps refer to Fish, PNAS, 2020 and Mancao, Blood, 2005.

  1. Thank you for your observation, both references were added to the sentence in the revised manuscript.
  2. Sections should stay on topic. This would also help with the massive number of references that are cited. A couple of examples:

Line 696: Reference #262, interesting but off topic.

Lines 712-713: Interesting, but off topic.

  1. The reference and the sentence were eliminated in the revised manuscript. In general the review was shortened to keep focus on the more important studies.
  2. The manuscript would substantially benefit from proof-reading for English language- particularly the introduction- as it is very difficult to digest in places. Words need to be corrected throughout the article to make the text more readable. Below are just a few examples:

Line 60: Via should be “route” in the sentence “EBV main via of transmission”.

Line 64: “released to the exterior”. Perhaps what is meant here is released into saliva?

Line 74 and 80: “latencies” should be latency programs.

Line 105: “bestowed severe immunosuppression” Please reconsider word choice here- bestow means “to gift or honor”.

Line 464: “EBV primo-infection” Perhaps primary infection is meant here?

Line 695: “increases de risk”

  1. Thank you very much for your comments. In addition to addressing the specific examples, we sent the manuscript for English proof-reading (see uploaded certificate).
